# Serum Protein Concentration and Serum Protein Fractions in Bottlenose Dolphins *(Tursiops truncatus)* under Human Care Using Agarose Gel Electrophoresis

**DOI:** 10.3390/ani13111745

**Published:** 2023-05-24

**Authors:** Federico Bonsembiante, Alessia Giordano, Claudia Gili, Sandro Mazzariol, Michele Berlanda, Carlo Guglielmini, Silvia Bedin, Maria Elena Gelain

**Affiliations:** 1Department of Animal Medicine, Production and Health, University of Padova, Viale Dell’Università 16, Legnaro, 35020 Padova, Italy; federico.bonsembiante@unipd.it (F.B.); michele.berlanda@unipd.it (M.B.); carlo.guglielmini@unipd.it (C.G.); silvia.bedin@unipd.it (S.B.); 2Department of Comparative Biomedicine and Food Science, University of Padova, Viale Dell’Università 16, Legnaro, 35020 Padova, Italy; sandro.mazzariol@unipd.it; 3Department of Veterinary Medicine and Animal Sciences, University of Milan, 26900 Lodi, Italy; alessia.giordano@unimi.it; 4Stazione Zoologica Anton Dohrn, Naples, Villa Comunale, 80121 Napoli, Italy; claudia.gili@szn.it

**Keywords:** marine mammals, agarose gel electrophoresis, reference intervals, albumin, serum globulins

## Abstract

**Simple Summary:**

Serum protein electrophoresis (SPE) is a widely used method to determine the concentration of a serum protein fraction in human and veterinary medicine. Serum protein electrophoresis allows the separation of serum proteins into several categories based on their ability to migrate in an electrical field on different supports. Typical migration patterns could be identified in the course of different disorders such as inflammation or neoplasia. However, to be clinically useful, the interpretation of SPE data had to be performed in light of an appropriate method—and species-specific reference intervals (RIs) and for all species, including marine mammals, attention should also be given to the different living environments. In this work, we established the RIs for serum protein fractions evaluated using agarose gel electrophoresis in bottlenose dolphins (*Tursiops truncatus*) under human care. These data are a new tool for health assessment in dolphins, thus, they may be useful for clinicians to better interpret laboratory data.

**Abstract:**

Serum protein electrophoresis (SPE) is the most used and reliable method to determine the percentage of serum protein subfractions. The interpretation of the kinetics of total proteins and albumin and globulin fractions is receiving increased attention in wild animals, as well as in domestic animals, due to the possibility of identifying typical pathologic patterns. However, the interpretation of these data had to be performed in light of an appropriate method—and species- specific reference intervals (RIs). In marine mammals, as well as other non-domestic species, specific attention should also be given to the different environment (free ranging vs. human managed) and the associated different exposure to environmental stimuli. The aim of this report was to establish RIs for the serum protein fractions evaluated using agarose gel electrophoresis (AGE) in bottlenose dolphins under human care. Peripheral blood samples were collected from 40 bottlenose dolphins during standard veterinary procedures to evaluate their health status. Total protein concentration was determined using the biuret method while AGE was performed using an automated system. A pooled dolphin’s serum sample was used to determine the intra-assay and inter-assay imprecision of AGE. The RIs were calculated using an Excel spreadsheet with the Reference Value Advisor set of macroinstructions. The intra and inter-assay imprecisions were 1.2% and 2.5%, respectively, for albumin; 2.9% and 5.7%, respectively, for α-globulins; 3.8% and 4.0%, respectively, for β-globulins; and 3.4% and 4.8%, respectively, for γ-globulins. The total protein, albumin, α-globulin, β-globulin, and γ-globulin concentrations were 65.5 ± 5.4 g/L, 45.5 ± 4.9 g/L, 8.0 ± 1.0 g/L, 5.0 ± 2.0 g/L, and 7.0 ± 2.0 g/L, respectively. We established the RIs for the total protein and serum protein fractions using AGE in bottlenose dolphins under human care.

## 1. Introduction

Serum protein electrophoresis (SPE) is the most used and trusted method in the separation of and quantification of serum proteins. Combined with hematological and biochemical profiles, it provides clinically useful information and it is an important tool for health screening. The interpretation of total proteins, and albumin and globulin fractions kinetics is receiving increased attention in both domestic and wild animals because typical pathologic distribution patterns characterize and discriminate among immune neoplastic or inflammatory disorders [1]. Serum protein electrophoresis allows the separation of the serum proteins into several categories based on their speed of migration, influenced by their size and superficial electric charge, in an electrical field after being stabilized on different supports (e.g., agarose gel). Thus, in most domestic mammal species, it is possible to identify six protein fractions characterized by the same migration speed: albumin and five globulins regions named α1, α2, β1, β2, and γ [1]. Cellulose acetate electrophoresis (CAE) and agarose gel electrophoresis (AGE) are the most used SPE methods in animals. Recently, veterinary laboratories have increasingly used capillary zone electrophoresis (CZE), an alternative to the gel-based methods’ electrophoresis, which allows a better peak resolution, and which is fully automated with a faster throughput [2,3]. However, the higher costs limit its widespread use in animals. In addition, a technique to identify a single-serum protein, namely high-resolution electrophoresis, was only validated in dogs among veterinary species [4] and is not widely available. The serum total protein evaluation is widely used as an essential tool to diagnose and monitor inflammatory diseases in cetacean species, such as killer whales [5], minke whales (*Balaenoptera acutorostrata*) [6], pantropical spotted dolphins (*Stenella attenuata*) [7], beluga (*Delphinapterus leucas*) [8], as well as other marine mammals, e.g., harbor seals (*Phoca vitulina*) [9] and walruses (*Odobenus rosmarus*) [10]. All these studies confirm that serum total protein analysis is one of the most used and commonly accepted markers of inflammation. Nevertheless, the clinical interpretation of these data had to be performed in light of an appropriate method—and species-specific reference intervals (RIs). Usually, the appropriate reference population should be selected based on different biological variables such as age, sex, and physiological state such as pregnancy [11]; moreover, in marine mammals, great consideration should also be given to the different environmental conditions (natural or artificial) since marine mammals under human care live in a controlled environment and thus are less exposed to pathogens.

The specific RIs of SPE are available for free-ranging bottlenose dolphins [12], but they cannot be applied to under human care bottlenose dolphins since the latter have lower total proteins and higher albumin levels [13]. These findings suggest that free ranging animals, which are more exposed to immunological stimuli, likely have a subclinical inflammatory status. This hypothesis is confirmed by the results of a recent paper [14] showing a higher level of some inflammatory markers in free-ranging bottlenose dolphins compared to the managed animals. The aim of this report was to establish the RIs for serum protein fractions evaluated using agarose gel electrophoresis (AGE) in bottlenose dolphins under human care.

## 2. Materials and Methods

Serum samples were obtained from 40 clinically healthy bottlenose dolphins (*Tursiops truncatus*) based on the history and normal results of physical examination, hematology, and serum biochemistry analyses. Dolphins were under human care in 4 different aquaria, 3 in Italy and 1 in Malta, among which there were 21 males and 19 females, and their median age was 18 years (minimum–maximum range: 1–51 years) (Table 1). Peripheral blood samples were obtained from each animal during routine veterinary procedures to evaluate their health status. The animals were fasted at the time of blood sampling to avoid lipemic serum samples that could affect the biochemical results. The animals were housed and handled in agreement with the Italian and Maltese Zoo directive law (DL 73/2005 and S.L.439.08, respectively) and all samples were obtained according to D.M. 469/2001, which establishes the management objectives and prescriptions to maintain the species *Tursiops truncatus* under human care. Blood was collected in K_3_-EDTA tubes for routine hematological analysis and in plain tubes to harvest serum for routine biochemical analyses and SPE; serum was obtained by the centrifugation of blood samples at 1500× *g* for 10 min. All samples were visually inspected in order to detect grossly hemolytic or lipemic serum; the samples were then stored at −20 °C until analysis.

### 2.1. Agarose Gel Electrophoresis

AGE was performed as previously described [15] using an automated system and kits provided by the manufacturer of the instrument (Hydrasis Sebia Italia Srl, Bagno a Ripoli, Firenze, Italy). Briefly, a 0.8% agarose gel was run in Trisbarbital buffer at pH 8.5, with migration time of 7 min at 36 Vh. Gels were stained with amido Schwarz, destained, and dried for scanning by the appropriate gel scanner. Data were then transferred to the software and visually inspected by a board-certified veterinary clinical pathologist to correct possible errors in a fraction separation automatically generated by the software (Phoresis, Sebia Italia Srl, Bagno a Ripoli, Firenze, Italy). The intra-assay repeatability (8 runs in the same day) and inter-assay repeatability (1 run/day for 8 working days) of AGE were assessed using a pooled dolphins’ serum sample, which enables the calculation of the coefficient of variation for each protein fraction (standard deviation/mean × 100).

### 2.2. Total Protein

The total protein (TP) concentration was determined by the biuret method on an automated spectrophotometer (Cobas Mira, Roche Diagnostics, Basel, Switzerland), and absolute values for each electrophoretic fraction were calculated based on the TP and percentage of the fraction.

### 2.3. Data Analysis

The RIs were obtained using an Excel spreadsheet with the Reference Value Advisor (v.2.1) set of macroinstructions, as reported by Geffré and colleagues [16]. The software performs the computations recommended by the International Federation of Clinical Chemistry-Clinical and Laboratory Standards Institute (CLSI) (2008) such as the descriptive statistics (mean, median, standard deviation (SD), minimum and maximum values), the tests of normality (Anderson–Darling with histograms and Q–Q plots and Box–Cox transformation), and the outlier analysis. Both the Dixon–Reed and Tukey tests were used to detect outliers: values were classified as “suspected” when they exceeded quartiles I or III minus or plus 1.5*the interquartile range (IQR) or as “far outliers” when they exceeded quartiles I or III minus or plus 3.0*IQR. Far outliers were removed and outliers classified as “suspected” were retained, as recommended by the ASVCP guidelines [17]. RIs were calculated using standard and robust methods on both non-transformed and transformed data. The software indicates the best method to define the RIs based on data distribution. A non-parametric bootstrap method was used to calculate the 90% confidence interval. The possible differences in the serum protein fractions depending on the sex and different age classes were investigated using a Mann–Whitney U test and Kruskal–Wallis test, respectively.

## 3. Results

Thirty-eight samples were included in this study (18 females and 20 males). Two samples, one from a female and one from a male, were excluded due to the marked hemolysis that affected the analysis of the electropherogram. Upon the visual inspection, four main protein fractions (albumin, α-globulins, β-globulins, and γ-globulins) were evident in all included animals (Figure 1). The pattern of all serum samples was interpreted as normal since no abnormal profile suggested that specific diseases were noted.

Intra- and inter-assay repeatability and mean percentage values ± standard deviation for each serum protein fraction are reported in Table 2.

The intra-assay imprecision ranged from 1.2% (albumin) to 3.8% (β-globulins), while the inter-assay imprecision ranged from 2.5% (albumin) to 5.7% (α-globulins).

In the whole dataset, the Tukey test identified only one outlier value among the γ-globulins percentage, which was removed. On the contrary, nine values identified as “suspected “outliers were retained.

Reference intervals for the total protein and serum protein electrophoretic fractions (percentage and concentration) based on AGE and biuret methods are presented in Table 3.

Based on data distribution, RIs were defined with BOX-COX transformed data for total proteins, α- and γ-globulins and total globulin concentrations, β-globulins percentage, and albumin/globulins ratio. The untransformed robust method was used for the albumin and γ-globulins percentage and the standard untransformed method for albumin and β-globulins concentrations and α-globulins percentage.

The results recorded from the female dolphins were not statistically different from those obtained in males for total protein, all serum protein subfraction (both percentages values and concentration) and albumin/globulin ratio (*p* values ranges from 0.32 to 0.98). Data partitioning by sex are presented in Table 4. The quartile distribution of cases based on age was as follows: 10 animals aged from 1 to 11 years (first quartile); 9 animals aged from 12 to 17 years (second quartile); 10 animals aged from 18 to 23 years (third quartile); and 9 animals aged from 29 to 51 years (fourth quartile). No significant differences were observed for any of the measured parameters, with the significance ranging from P = 0.068 for albumin g/L to P = 0.81 for alpha-globulin g/L.

Box-plot and histograms of the data distribution are presented in Figure 2 and Figure 3, respectively.

## 4. Discussion

The RIs for the serum protein fractions in AGE, reported in the present study, extend the toolbox of diagnostic and monitoring tools available for bottlenose dolphins.

Serum protein electrophoresis is widely used in veterinary medicine to quantify albumin and globulin fractions and identify their changes in concentration, which could be supportive of the inflammatory status or indicative of infectious diseases [18]. The interpretation of total proteins and albumin and globulin fractions kinetics is also receiving increased attention in marine mammals in which, as in terrestrial mammals, typical pathologic patterns can be identified in several disorders, such as inflammatory diseases [19]. Specifically, the increased globulin concentration is often suggestive of increased acute phase protein (APP) levels, since, in mammals, α globulins include α1–acid glycoprotein and haptoglobin, while transferrin and serum amyloid-A migrate in β-globulins and IgG and C-reactive protein in γ-globulins [1,20]. Although SPE does not allow the identification of the amount of a single protein (except for albumin), but rather the amount of a group of proteins with the same electrophoretic mobility [21], it is a valued tool in routine health assessment and clinical settings. In fact, SPE is a relatively inexpensive method of rapidly identifying the state of a disease and, in some cases, such as the rescue of stranded marine mammals, it can help assess the clinical state of animals, and assess the response to treatment and rehabilitation before their release into the wild. It is also noted that, in recent years, wild marine mammals were threatened by pathogens such as *Morbillivirus*, *Brucella ceti*, and *Toxoplasma gondii* [22]: the assessment of serum protein fractions and the possible identification of a specific pattern could increase the knowledge of host-specific pathogen interactions. This is even more likely considering that the reagents for the most commonly used APPs, which are more sensitive markers of inflammation, do not cross-react with all marine mammals’ proteins, and specific APP reagents for these species are not easily available [23].

The measurement of serum protein fractions can be performed with different electrophoretic techniques, such as CAE, AGE, and CZE. Over the years, various techniques have been developed to increase the identification and separation of single proteins, such as high-resolution electrophoresis, a technique that has been validated in dogs (Abate et al., 2000) but has never been applied to marine mammals. Nowadays, CZE has replaced classical AGE in humans, due to its higher resolution [24]. In both human and veterinary medicine, the high resolution of CZE allows one to decrease the coefficient of variation and increase the sensitivity to detect protein subfractions [15,25]. These features lead to a higher probability of observing unusual SPE profiles, mainly in non-domestic species [18], or to detect “abnormal” profiles, as in the recently reported case of hereditary bisalbuminemia in bottlenose dolphins [13,26]. Overall, the differences in precision and resolution between different techniques such as AGE and CZE necessitate the generation of method-specific RIs. Recently, the reference intervals for serum protein fractions obtained with CZE in free-ranging bottlenose dolphins were published [27], and the authors observed a lower level of albumin compared to the data previously published from dolphins under human care [13]. Similarly, Flower and colleagues (2020) [14] reported a significantly higher concentration of several inflammatory markers in free-ranging dolphins compared to those from animals under professional care, confirming that environmental challenges may stimulate the immune response in free-ranging dolphins. For these reasons, the interpretation of clinical-pathological data needs appropriate RIs, not only based on species and method, but also related to the living environment.

Despite its advantages, the higher cost of CZE, compared to most traditional techniques, hampered its wide diffusion in veterinary laboratories as well as its use in non-domestic animals. Thus, we decided to determine the RIs of serum protein fractions in bottlenose dolphins under human care using AGE because it is the most commonly used method for measuring serum protein fractions in marine mammals [12]. From an analytical point of view, the intra-assay imprecision for AGE in bottlenose dolphins was between 1.2% and 3.8% and the inter-assay imprecision was between 2.5% and 5.7% for the different protein fractions. Our data are similar to the intra- and inter-assay imprecisions recorded for AGE in canine, feline, and rat serum samples [15,28], and are noticeably lower compared to the intra-assay imprecision reported by Zaias and colleagues for AGE in bottlenose dolphins [27]. Thus, even if it is possible that the variability recorded in this study influenced the RIs of the less represented serum protein fractions, as can also happen for other species, the imprecision recorded in the present study can be considered optimal.

The concentrations of TP, α-globulins, and γ-globulins in our samples were slightly lower compared to the previously published data in free-ranging bottlenose dolphins [12]—while the concentration of albumin and the albumin/globulins ratio was slightly higher. Moreover, the RIs obtained with CZE in another group of free-ranging bottlenose dolphins [27] showed a slightly higher concentration of TP with a significantly lower A/G ratio, in addition to higher β-globulins and γ-globulins levels compared to our results. The same trend was also confirmed by comparing our data with those published by Flower and colleagues (2020) [14]: while the concentration of serum protein fractions obtained in apparently healthy dolphins under professional care overlaps with our data, the values of free-ranging animals showed a sort of pro-inflammatory status. Acute phase proteins such as haptoglobin, α1-antitrypsin, α1-antichymotripsin, and α2-macroglobulin migrate in the α-globulins fraction, and IgG and CRP migrate in the γ-globulins fraction, while albumin acts as a negative acute phase protein [1]. The lower concentration of “inflammatory’’ proteins associated with a higher concentration of albumin and the consequent higher albumin/globulins ratio reported in our study population could thus reflect lower antigenic stimulation in the animals housed in aquaria compared to the free-ranging populations. Moreover, in the paper by Schwacke and colleagues (2009) [12] on free-ranging bottlenose dolphins, the inclusion of possibly non-healthy individuals was not excluded, so it is possible that the RIs proposed did not represent a strictly disease-free population. On the contrary, the serum samples used in our work were collected from animals with a normal clinical history and physical examination.

In addition to the environment, it is well documented that sex, age, and season also influence the clinical-pathologic data, including the serum protein distribution in non-domestic species such as reptiles and birds [29,30,31]. Our data show no statistically significant differences between the subfraction of serum protein in female and male animals. This result, which has been linked to a small number of animals of each sex (18 females and 20 males), did not allow us to calculate sex-specific RIs.

The current study is limited by the relatively small sample size that hampered the possibility of further partitioning RIs also based on age. In studies evaluating non-domestic animals, it is difficult to enroll 120 reference individuals considering the optimal number for establishing RIs (Clinical and Laboratory Standards Institute 2008, [16]). However, we included 38 reference individuals, a number that is considered acceptable for calculating RIs [17]. These intervals, alongside other already published data, can be used as a guide for interpreting the hematological and biochemical values in this species. An alternative and promising approach in the case of a few reference animals (i.e., the animals used to calculate RIs) is to create individualized RIs (iRIs). However, according to Freeman and his colleagues [32], for the creation of iRIs, each reference animal must be sampled weekly for 4–6 weeks. However, in our study, we used the remaining serum aliquots collected for health checks and the samples were not specifically obtained for the RI generation, so this approach was impossible to achieve.

## 5. Conclusions

In this study, we reported RIs for serum protein fractions obtained with AGE in bottlenose dolphins maintained in a controlled environment under human care, as an additional tool for assessing the health of these animals. These data, even if obtained in a small number of animals which was restricted due to the limited availability of this species under human care, could represent a valid tool for clinicians to better interpret laboratory data.

Further studies, with larger available sample sets, will support the development of reference intervals for bottlenose dolphins under human care partitioned by age and sex.

## Figures and Tables

**Figure 1 animals-13-01745-f001:**
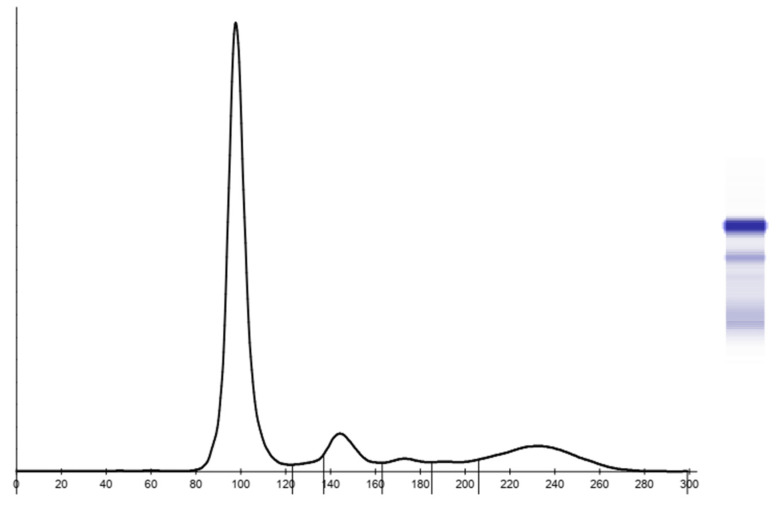
Example of agarose gel serum protein electropherogram in bottlenose dolphins.

**Figure 2 animals-13-01745-f002:**
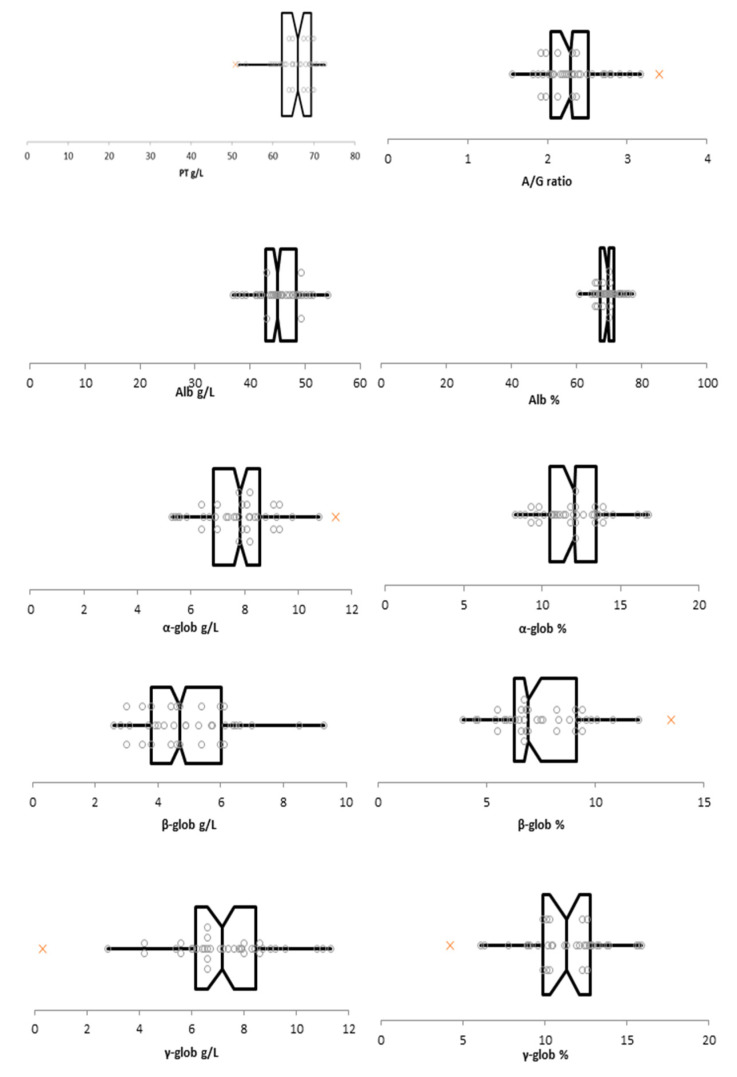
Box-plot of data for the total protein, albumin/globulins ratio, albumin (g/L and %) α-globulins (g/L and %), β-globulins (g/L and %), and γ-globulins (g/L and %). The box represents the interquartile range (IQR) defined by the 25th (Q1) and 75th (Q3) percentiles with the vertical line representing the median. The horizontal lines are the limits of outliers’ distribution according to Tukey rule. Suspected outliers are indicated with orange x. Abbreviations: PT: total protein; A/G ratio: albumin-to-globulin ratio; Alb: albumin; α-glob: α-globulins; β-glob: β-globulins; γ-glob: γ-globulins.

**Figure 3 animals-13-01745-f003:**
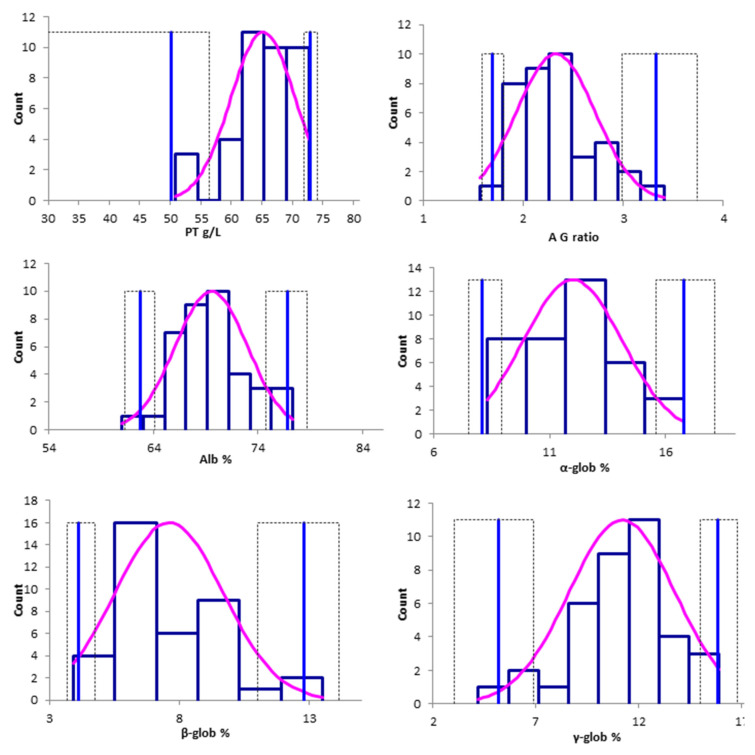
Histograms of the untransformed data distribution for the total protein (g/L), albumin/globulins ratio, albumin (%) α-globulins (%), β-globulins (%), and γ-globulins (%). The blue bars show the relative frequency of each value, the pink lines represent the fitted distribution. The vertical light blue lines indicate the upper and lower limit of the reference limits, and the dashed lines indicate the 90% CIs of each limit. Abbreviations. PT: total protein; A/G ratio: albumin-to-globulin ratio; Alb: albumin; α-glob: α-globulins; β-glob: β-globulins; γ-glob: γ-globulins.

**Table 1 animals-13-01745-t001:** Age and sex of the bottlenose dolphins included in the study. Abbreviations. f: female; m: male; n.a.: not assessed.

Animal	Age (Years)	Sex
1	36	f
2	1	f
3	32	f
4	20	f
5	29	f
6	38	f
7	6	f
8	13	f
9	18	f
10	19	f
11	17	f
12	35	f
13	29	f
14	17	f
15	20	f
16	51	f
17	2	f
18	12	f
19	12	f
20	17	m
21	31	m
22	22	m
23	8	m
24	11	m
25	18	m
26	5	m
27	5	m
28	4	m
29	n.a.	m
30	n.a.	m
31	12	m
32	21	m
33	1	m
34	18	m
35	22	m
36	16	m
37	15	m
38	4	m
39	23	m
40	41	m

**Table 2 animals-13-01745-t002:** Intra-assay (8 runs) and inter-assay (8 runs) imprecision, expressed as a coefficient of variation (CV) recorded for the AGE of the bottlenose dolphin pooled serum. For each fraction, the mean percentage ± SD are reported between brackets.

	Intra-Assay CV (%)	Inter-Assay CV (%)
Albumin	1.2 (60.9 ± 0.73)	2.5 (64.13 ± 1.6)
α-globulins	2.9 (14.1 ± 0.4)	5.7 (12.0 ± 0.7)
β-globulins	3.8 (9.1 ± 0.4)	4.0 (8.6 ± 0.3)
γ-globulins	3.4 (15.9 ± 0.5)	4.8 (15.4 ± 0.7)

**Table 3 animals-13-01745-t003:** Reference intervals for the total protein and electrophoretic fractions’ concentration and percentage in bottlenose dolphins. Abbreviations. TP: total protein; SD: standard deviation; LRL: lower reference limit; URL: upper reference limit; CI: confidence interval; Out: outliers; S: suspected; R: removed.

	Mean	Median	SD	Min	Max	LRL 90% CI	URL 90% CI	Out	Distribution
TP (g/L)	65.0	67.0	5.0	51.0	74.0	54.0 (36.0–57.0)	74.0 (72.0–75.0)	2S	BOX-COX transformed
Albumin (g/L)	45.0	45.0	4.0	37.0	54.0	37.0 (35.0–39.0)	53.0 (52.0–55.0)	0	Untransformed
α-globulins (g/L)	8.0	8.0	1.0	5.0	11.0	5.0 (5.0–6.0)	11.0 (10.0–12.0)	1S	BOX-COX transformed
β-globulins (g/L)	5.0	5.0	2.0	3.0	9.0	2.0 (1.0–3.0)	8.0 (7.0–9.0)	0	Untransformed
γ-globulins (g/L)	7.0	7.0	2.0	0.0	11.0	2.0 (1.0–4.0)	11.0 (10.0–12.0)	1S	BOX-COX transformed
Albumin (%)	69.5	69.6	3.4	61	77.3	62.3 (60.8–63.7)	76.4 (74.5–78.2)	1S	Untransformed robust method
α-globulins (%)	12.0	12.1	2.1	8.3	16.8	7.6 (6.6–8.5)	16.4 (15.3–17.4)	0	Untransformed
β-globulins (%)	7.6	6.9	2.1	3.9	13.5	4.2 (3.7–4.7)	12.6 (11.3–14.4)	1S	BOX-COX transformed
γ-globulins (%)	11.2	11.3	2.6	4.2	15.9	6.1 (4.9–7.5)	16.8 (15.4–18)	1R, 1S	Untransformed robust method
Total globulins (g/L)	19.9	20.0	3.1	12.2	26.8	13.4 (11.7–15.1)	25.8 (24.5–27.1)	1S	BOX-COX transformed
Albumin /globulins ratio	2.3	2.3	0.4	1.6	3.4	1.7 (1.6–1.8)	3.3 (3.0–3.7)	1S	BOX-COX transformed

**Table 4 animals-13-01745-t004:** Results for the total protein and electrophoretic fractions concentration and percentage in 18 females and 20 males bottlenose dolphins. No statistically significant differences were identified by the Mann–Whitney U test. Abbreviations. TP: total protein; SD: standard deviation.

	Female (18)				Male (20)			
	Mean ± SD	Median	Min	Max	Mean ± SD	Median	Min	Max
TP (g/L)	65.77 ± 4.89	67.30	53.40	72.00	64.66 ± 5.67	64.85	50.80	72.70
Albumin	45.70 ± 4.26	45.36	37.60	54.10	44.79 ± 3.63	44.74	36.90	50.20
(g/L)								
α-globulins	7.51 ± 1.19	7.80	5.40	9.20	8.06 ± 1.56	7.85	5.30	11.40
(g/L)								
β-globulins	5.06 ± 1.72	4.65	2.80	9.30	4.87 ± 1.32	4.78	2.60	7.00
(g/L)								
γ-globulins	7.37 ± 1.69	7.30	4.20	11.00	6.93 ± 2.49	6.65	0.30	11.30
(g/L)								
Albumin	69.65 ± 3.85	69.75	61.00	77.30	69.40 ± 3.10	68.80	64.60	76.00
(%)								
α-globulins	11.47 ± 1.89	11.95	8.30	13.90	12.46 ± 2.25	12.05	9.30	16.80
(%)								
β-globulins	7.65 ± 2.30	7.05	4.50	13.50	7.54 ± 1.91	6.90	3.90	10.80
(%)								
γ-globulins	11.25 ± 2.41	11.25	6.1	15.9	11.11 ± 2.79	12.30	4.20	15.60
(%)								
Total globulins	19.95 ± 3.00	19.1	15.1	26.8	19.87 ± 3.08	20.35	12.2	23.97
(g/dL)								
Albumin/globulins ratio	2.34 ± 0.43	2.30	1.56	3.40	2.30 ± 0.36	2.20	1.82	3.16

## Data Availability

Raw data are available by writing an email to the corresponding author.

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
