# Peer review of "Serum Protein Concentration and Serum Protein Fractions in Bottlenose Dolphins (Tursiops truncatus) under Human Care Using Agarose Gel Electrophoresis"

_animals, 2023, doi:10.3390/ani13111745_

Round 1

Reviewer 1 Report

SPE is a widely used method to determine the concentration of serum protein fraction in human and veterinary medicine. However, the authors should do more before submission, such as: 1. analysis on possible causes of the outliers with the other blood work values and clinical findings although they were supposed to be clinically healthy; 2. the application of the RIs on some clinical cases for demonstrating the diagnosis values.

Author Response

  1. In our data set, only one value was excluded as considered a far outlier: it was the percentage value of gamma globulin in animals 33 (see Table 1), a very young male. The absolute values of gamma-globulin in this animals was a “suspected” outliers and no other values of these animals were considered far o suspected outliers. Other suspected outliers values (9 observations out of a total of 418) were maintained, as recommended by the ASVCP guidelines. The presence of suspected outliers in otherwise healthy animals ( as specified in M&M section, other haematological analysis were normal) is not always a sign of a possible subclinical disease, but it could be related to normal biological variation. For these reasons, an alternative approach in the case of a few reference animals is the individual reference intervals (see also response to review 3), but unfortunately we have not the possibility of using multiple samples from the same animal.
  2. It is undeniable that the diagnostic value of RI is evident when applied to clinical environments, but the application of RI generated on diseased animals is outside the scope of our research. Our aim was to create a toll that clinicians could use to interpret laboratory data daily. Because the number of animals in non-domestic species, whether free-ranging or managed, is limited, we believe that all possible clinical-pathological data on healthy animals can be used to help assess the health of these species.

Reviewer 2 Report

Dear Authors,
congratulations on this work which I found very well-designed and developed.

Please, find below some suggestions which could improve the quality and scientific soundness of the work.

1. Due to the importance of the proposed innovative tool for health assessment and the selection of wild species for this study, despite the under human care conditions, it would be appropriate to extend the discussion or mention in the conclusion the conservation aspects and applications of this work in free-ranging specimens.

2. Because the M&M mentions the sex of the dolphins, it would be interesting if the Authors could just mention in the Results and Discussion section the relevant results on the difference between sexes (if any or not). 

3. Line 83: It would be better to define the condition of the animals as follows: "[...] in bottlenose dolphins under human care."

4. Line 87: Same comment as in Line 83.

5. Line 216: Same comment as in Line 83.

6. Line 226: Same comment as in Line 83.

7. Line 268: Same as in Line 83.

Author Response

  1. Due to the importance of the proposed innovative tool for health assessment and the selection of wild species for this study, despite the under human care conditions, it would be appropriate to extend the discussion or mention in the conclusion the conservation aspects and applications of this work in free-ranging specimens.

Thank you to the reviewer for this suggestion. We add some observations on these subjects to the discussion.

  1. Because the M&M mentions the sex of the dolphins, it would be interesting if the Authors could just mention in the Results and Discussion section the relevant results on the difference between sexes (if any or not). 

We added in the manuscript the data from females and males dolphins. We also evaluate the possible influences of sex, but none statistically significantly, differences were found for any of the parameters analyzed. We added this information in the text and the results in a new table.

  1. Line 83: It would be better to define the condition of the animals as follows: "[...] in bottlenose dolphins under human care."
  2. Line 87: Same comment as in Line 83.
  3. Line 216: Same comment as in Line 83.
  4. Line 226: Same comment as in Line 83.
  5. Line 268: Same as in Line 83.

We changed the manuscript according with all the review’s suggestions

Reviewer 3 Report

The work is very interesting as the diagnostic management for these animals is still little known. The study allows, in fact, to increase the knowledge on the diagnostic resources and the reference ranges of marine mammals for the evaluation of their health status, especially for animals managed in a controlled environment under human care.

However, some questions arise. Authors should consider the following comments.

1) Indicate whether the samples were taken on an empty stomach.

2) Why was only one sample taken for animals?  It would have been useful to repeat the sampling at a distance of time to monitor the trend and the reference ranges of the parameters analysed.

Author Response

1) Indicate whether the samples were taken on an empty stomach.

Thanks to the reviewer. The animals were fasted at the time of sampling. We added this information in the M&M section

2) Why was only one sample taken for animals?  It would have been useful to repeat the sampling at a distance of time to monitor the trend and the reference ranges of the parameters analysed.

The approach suggested by the reviewer is generally used for the calculation of Individual Reference Intervals (IRi). The creation of IRi is a new and promising approach, especially for non-domestic species in which it is difficult to have a large number of reference individuals to create the reference intervals. However to create the individual reference interval, the animals must be sampled weekly for a period of 4-6 weeks. In our study we used only one sample because we used residual serum aliquots collected for health checks. We added this information in the text.

Reviewer 4 Report

Comments to the Author

General comments:

This study established the RIs for total protein and serum protein fractions using AGE in managed bottlenose dolphins.

The manuscript could be considered for publication after being minor revisions. Some information needs to be provided as follows.

Specific comments:

1. In lines 104, Data where then transferred to, should be revised to “Data were then transferred to”.

2. In lines 107, y (1 run/day for 8 working day)

  for 8 working day— for 8 working days

3.    In lines 157, standard untransformed method.

the standard untransformed method

4.    In lines 181, The blue bars show the relative frequency of each values, the pink lines represent the fitted distribution.

The blue bars show the relative frequency of each value, and the pink lines represent the fitted distribution.

5.    In lines 181, This is even more likely considering that reagents for the most common used APPs

common--commonly

6. In lines 213,    Overall, the differences in precision and resolution between different techniques such as AGE and CZE makes necessary the generation of method-specific RIs.

makes--make

7. In lines 217, Similarly, Flower and colleagues (2020) [13] reported a significant higher

significant—significantly

8.    In lines 231, and noticeable lower compared to intraassay imprecision reported by Zaias and colleagues for AGE in bottlenose dolphins

noticeable—noticeably

9.    In lines 217, while the concentration of albumin and the albumin/globulins ratio were slightly higher.

were—was

10.  In lines 240, with a significant lower A/G ratio, 240 higher β-globulins and γ-globulins levels compared to our results.

significant—significantly

Author Response

We changed the manuscript according with all the review’s suggestions

Round 2

Reviewer 1 Report

The authors used data from dozens of clinically healthy dolphins at different ages. However, no analysis of the effect on the data from the variables such as ages and medical history. Moreover, it is not appropriate that the outliers were just regarded as normal variation.  It would be better if the authors can show the presumed RIs could be used for identified abnormal health conditions.  Therefore, I suggest reject the present form and would reconsider it if the authors can provide the revised manuscript following the reviewers' comments.

Author Response

The authors used data from dozens of clinically healthy dolphins at different ages. However, no analysis of the effect on the data from the variables such as ages and medical history.

Our dataset is not sufficiently large to conduct a meaningful analysis of age distribution. However, in response to the reviewer's request, we analysed the data based on this variable. According to the age classes proposed by Venn-Watson and colleagues (“Effect of age and sex on clinicopathologic reference ranges in a healthy managed Atlantic bottlenose dolphin population”), our population was composed by 7 calf dolphins, 2 juvenile dolphins, 21 adult dolphins and 6 geriatric dolphins (for 2 dolphins we don’t have information regarding the age). This stratification (the overrepresentation of adult dolphins) hampered the statistical analysis. Thus, we divided the animals in quartile and the distribution of our cases is as follows: 10 animals in the first quartile (aged 1 to 11 years), 9 animals in the second quartile (aged 12 to 17 years), 10 animals in the third quartile (aged 18 to 23 years), and 9 animals in the fourth quartile (aged 29 to 51 years). No significant differences were observed for any of the measured parameters, with significance ranging from P=0.068 for albumin g/L to P=0.81 for alpha-globulin g/L. We have included this analysis in the text. As previously stated, the animals were clinically healthy, thus precluding any analysis based on their clinical history. 

Moreover, it is not appropriate that the outliers were just regarded as normal variation

We handled the outliers as suggested by international guidelines, for which we have already provided bibliographic references, and as recommended by the creators of the software used (for which bibliographic references are also present in the manuscript), which is widely used in veterinary medicine studies. We removed one data point identified as a true outlier, while the "suspected" outliers were retained.

We understand that the reviewer may handle outliers differently, but we believe it is more appropriate to adhere to the published guidelines.

It would be better if the authors can show the presumed RIs could be used for identified abnormal health conditions

As has already been stated, even if interesting, using our data to identify pathological situations in clinical environments is not within the scope of our work. Our work belongs to the field of research whose main objective is to establish reference intervals for the population, taking into account a reference population (in our case, animals under human care) without pathology. As the reviewer knows, many studies have determined the electrophoresis RIs for various species (see some examples below). As explained in the previous response, although we are aware of the limited number of animals tested in our study, we are also convinced that, given the limited possibility of having numerous controlled samples of these animals, our work could be considered an additional tool available to veterinarians treating dolphins.

Here, some articles in which RIs for electrophoresis of serum protein were calculated on healthy animals ( non-domestic and domestic) and were not compared to animals with diseases:

  1. Rosa, S.; Silvestre-Ferreira, A.C.; Sargo, R.; Silva, F.; Queiroga, F.L. Hematology, Biochemistry, and Protein Electrophoresis Reference Intervals of Western European Hedgehog (Erinaceus europaeus) from a Rehabilitation Center in Northern Portugal. Animals 2023, 13, 1009.
  2. Shinder, S.L.; Cray, C.; Hammerschlag, N.; Merly, L. Serum Protein Electrophoresis Reference Intervals for Six Species of Wild-Sampled Sharks in South Florida. Integr Comp Biol. 2022, 62, 1547–1556.
  3. Jaensch, S.; Howard, J.G. Establishment of Reference Intervals for Plasma Protein Electrophoresis and Comparison of Biochemical and Protein Electrophoresis Evaluation of Albumin in Central Bearded Dragons (Pogona vitticeps). Aust. Vet. J. 2022, 100, 446–450.
  4. Chagas, C.R.F.; Lima, C.F. da M.; Hany Lima Gonzalez, I.; Borges Salgado, P.A.; Monticelli, C.; Ramos, P.L. Hematologic and Biochemical Reference Intervals of Brown-Throated Sloths (Bradypus variegatus). Vet. Clin. Pathol. 2022, 51, 126–133.
  5. Morón-Elorza, P.; Rojo-Solís, C.; Steyrer, C.; Álvaro-Álvarez, T.; Valls-Torres, M.; Ortega, J.; Encinas, T.; García-Párraga, D. Increasing the Data on Elasmobranch Plasma Protein Electrophoresis: Electrophoretogram Reference Values Determination in the Undulate Skate (Raja undulata) and the Nursehound Shark (Scyliorhinus stellaris) Maintained under Human Care. BMC Vet. Res. 2022, 18, 380.
  6. Kneeland, M.; Berman, E.; Grade, T.; Cooley, J.; Vogel, H.; Schoch, N.; Cray, C.; Stout, V.; Evers, D.; Pokras, M. Plasma Biochemistry And Protein Electrophoresis Reference Intervals Of The Common Loon (Gavia immer). J. Zoo Wildl. Med. 2020, 561–570.
  7. Rosenberg, J.F.; Wellehan, J.F.X.; Crevasse, S.E.; Cray, C.; Stacy, N.I. Reference Intervals for Erythrocyte Sedimentation Rate, Lactate, Fibrinogen, Hematology, and Plasma Protein Electrophoresis in Clinically Healthy Captive Gopher Tortoises (Gopherus polyphemus). J. Zoo Wildl. Med. 2018, 49, 520–527.
  8. Wernick, M.B.; Martin-Jurado, O.; Beaufrère, H.; Howard, J.; Samour, J. Serum Protein Electrophoresis Reference Values in the Gyrfalcon (Falco rusticolus). Comp. Clin. Path. 2018, 27, 493–497.
  9. Miglio, A.; Antognoni, M.T.; Maresca, C.; Moncada, C.; Riondato, F.; Scoccia, E.; Mangili, V. Serum Protein Concentration And Protein Fractions In Clinically Healthy lacaune And Sarda Sheep Using Agarose Gel Electrophoresis. Vet. Clin. Pathol. 2015, 44, 564–569.
  10. Musilová, A.; Knotková, Z.; Pinterová, K.; Knotek, Z. Variations of Plasma Protein Electrophoresis in Healthy Captive Green Iguanas (Iguana iguana). Vet. Clin. Pathol. 2015, 44, 243–248.
  11. Flower, J.E.; Byrd, J.; Cray, C.; Allender, M.C. Plasma Electrophoretic Profiles and Hemoglobin Binding Protein Reference Intervals in the Eastern Box Turtle (Terrapene carolina carolina) and Influences of Age, Sex, Season, and Location. J. Zoo Wildl. Med. 2014, 45, 836–842.
  12. Alberghina, D.; Giannetto, C.; Vazzana, I.; Ferrantelli, V.; Piccione, G. Reference Intervals for Total Protein Concentration, Serum Protein Fractions, and Albumin/Globulin Ratios in Clinically Healthy Dairy Cows. J. Vet. Diagnostic Investig. 2011, 23, 111–114.
  13. Dawson, D.R.; Defrancisco, R.J.; Mix, S.D.; Stokol, T. Reference Intervals for Biochemical Analytes in Serum and Heparinized Plasma and Serum Protein Fractions in Adult Alpacas (Vicugna pacos). Vet. Clin. Pathol. 2011, 40, 538–548.
  14. Spagnolo, V.; Crippa, V.; Marzia, A.; Sartorelli, P. Reference Intervals for Hematologic and Biochemical Constituents and Protein Electrophoretic Fractions in Captive Common Buzzards (Buteo buteo). Vet. Clin. Pathol. 2006, 35, 82–87.
  15. Fayos, M.; Couto, C.G.; Iazbik, M.C.; Wellman, M.L. Serum Protein Electrophoresis In Retired Racing Greyhounds. Vet. Clin. Pathol. 2005, 34, 397–400.
